# RWT/SaVR—A Simple and Highly Accurate Measure Screening for Transthyretin Cardiac Amyloidosis

**DOI:** 10.3390/jcm11144120

**Published:** 2022-07-15

**Authors:** Elsa Arnberg, Per Eldhagen, Viktor Löfbacka, Ashwin Venkateshvaran, Björn Pilebro, Per Lindqvist

**Affiliations:** 1Department of Clinical Physiology, Surgical & Perioperative Sciences, Umea University, SE-901 85 Umea, Sweden; viktor.lofbacka@gmail.com (V.L.); per.lindqvist@umu.se (P.L.); 2Cardiology Unit, Department of Medicine, Karolinska Institute, SE-171 77 Stockholm, Sweden; per.eldhagen@regionstockholm.se (P.E.); ashwinvenkateshvaran@gmail.com (A.V.); 3Department of Cardiology, Public Health and Clinical Medicine, Umea University, SE-901 85 Umea, Sweden; bjorn.pilebro@regionvasterbotten.se

**Keywords:** cardiac amyloidosis, ECG, left ventricular hypertrophy, relative wall thickness, transthyretin

## Abstract

Background: Cardiac amyloidosis is an underdiagnosed condition and simple methods for accurate diagnosis are warranted. We aimed to validate a novel, dual-modality approach to identify transthyretin cardiac amyloidosis (ATTR-CA), employing echocardiographic relative wall thickness (RWT), and ECG S-wave from aVR (SaVR), and compare its accuracy with conventional echocardiographic approaches. Material and methods: We investigated 102 patients with ATTR-CA and 65 patients with left ventricular hypertrophy (LVH), all with septal thickness > 14 mm. We validated the accuracy of echocardiographic measures, including RWT, RWT/SaVR, posterior wall thickness (PWT), LV mass index (LVMI), left atrial volume index (LAVI), global longitudinal strain (GLS), and relative apical sparing (RELAPS) to identify ATTR-CA diagnosed using DPD-scintigraphy or abdominal fat biopsy. Results: PWT, RWT, RELAPS, troponin, and RWT/SaVR were significantly higher in ATTR-CA compared to LVH. RWT/SaVR > 0.7 was the most accurate parameter to identify ATTR-CA (sensitivity 97%, specificity 90% and accuracy 91%). RELAPS was found to have much less accuracy (sensitivity 74%, specificity 76% and accuracy 73%). Conclusion: We can confirm the very strong diagnostic accuracy of RWT/SaVR to identify ATTR-CA in patients with septal thickness > 14 mm. Given its high sensitivity and specificity, RWT/SaVR > 0.7 has the potential to implement as a non-invasive, simple, and widely available diagnostic tool when screening for ATTR-CA.

## 1. Introduction

Transthyretin cardiac amyloidosis (ATTR-CA) is a disease caused by misfolded transthyretin-protein (TTR). Misfolded TTR aggregates and accumulates in the heart muscle. This causes myocardial cell death and increasing thickening of the heart, which interferes with the myocardial function and can later cause heart failure (HF). ATTR is either inherited (ATTRv) or acquired (ATTRwt) and both types may have similar cardiac involvement [1]. HF due to cardiac amyloidosis (CA) is often clinically misinterpreted as hypertrophy-related HF. Hypertrophy-related HF arise from various cardiovascular conditions with increased intraventricular pressure and myocardial workload, i.e., aortic stenosis and hypertension or factors intrinsic to cardiac myocytes in sarcomeric hypertrophic cardiomyopathy (HCM) [2]. The pathophysiological mechanisms in ATTR-CA are fundamentally different when compared to classical hypertrophy-related heart failure, and treatment strategies differ [3].

Since ATTR-CA has a unique pathophysiological pattern, is it important to avoid common and conventional HF pharmaceuticals. ß-blockers, ACEI/ARB, and diuretics are commonly used to treat HF. ß-blockers inhibit the cardiovascular sympathetic activity and thereby block the full inotropic and chronotropic potential of the heart. Amyloid hearts require adequate heart rate in order to fulfill the peripheral blood need. ACEI/ARB are usually poorly tolerated in ATTR-CA due to the risk of vasodilatation, causing symptomatic hypotension with disease worsening. However, diuretic treatment is well tolerated and is also an important treatment option in ATTR-CA patients [4,5].

In addition, several specific ATTR-CA treatment options are now available or are being studied in late clinical phase trials. Available treatments either stabilize circulating TTR [6] or decrease the hepatic TTR-production [7,8].

Differentiating ATTR-CA from LVH is complex since they share common phenotype characteristics. The signature appearance of both conditions is abnormal myocardial thickness. There are, however, some clues to differentiate between the conditions. Myocardial hypertrophy is mostly pronounced in the interventricular septum. In comparison, TTR accumulation is more generally distributed, with concentric increased wall thickness, but with less influence in the apical section. This generates a certain heart dynamic that can be illustrated with an echocardiographic presentation called apical sparing pattern [9]. Commonly, ATTR-CA also develops increased biventricular wall thickness, [10] decreased long axis function and sometimes restrictive filling pressures [11]. However, this might also be found in LVH patients.

Differentiation between LVH and ATTR-CA can also be completed with ECG-parameters, especially in combination with echocardiography [12]. Hypertrophic hearts commonly generate more pronounced electrical activity and thereby higher QRS amplitudes in ECG. TTR amyloid infiltration does not have this characteristic but rather normal to lower voltage, which, in combination with increased wall thickness, raises suspicion of CA.

Despite increasing awareness, ATTR-CA continues to be an underdiagnosed cause of HF [13]. We and others have presented different methods to improve diagnosis strategy. We have found high diagnostic accuracy for differentiating ATTR-CA from HF controls by using a ratio of relative wall thickness from echocardiography and S amplitude in aVR from ECG [14].

The aim of this study is to validate the accuracy of the RWT/SaVR formula in a large cohort of ATTR-CA and LVH patients.

## 2. Materials and Methods in the Deviation and Validation Groups

This was a retrospective case control study, comparing ATTR-CA with a heterogenous group of LVH patients. Diagnosis was set at the Umea University Hospital in Umea, Sweden. This study also included a validation cohort comprising patients with ATTR-CA from Karolinska University Hospital in Stockholm, Sweden.

### 2.1. ATTR-CA Population

In total, 102 patients with ATTR-CA and interventricular septum > 14 mm were retrospectively identified from patients seen at Umea University Hospital. ATTR-CA diagnosis was set either non-invasively according to the algorithm proposed by Gillmore et al. using DPD scintigraphy [15] or by abdominal fat biopsy [16]. Patients with a DPD-scintigraphy uptake Perugini grade 2 or 3 were further analyzed with TTR-gene sequencing to differentiate between ATTRwt and ATTRv. A genetic workup (sequencing of the TTR gene) was performed for all patients with a DPD scintigraphy uptake of Perugini grade 2 or 3, in order to differentiate between ATTRwt and ATTRv [17]. Overall, 48 of these patients had previously been part of a recent publication [14].

The exclusion of light chain amyloidosis (AL-amyloidosis) was performed using blood and urine samples, which were analyzed for serum free light chain (FLC) abnormalities (Freelite, Binding Site reagent, reference range 0.27–1.64) and the presence of monoclonal bands. Patients with abnormalities in these analyses were evaluated, and their clinical history and disease progression were reviewed to assess the probability of AL-amyloidosis.

### 2.2. LVH Population

A total of 65 patients were included in an LVH control group, all with septal thickness > 14 mm. This group included 14 patients with severe aortic stenosis, 24 with HCM, and 27 with mainly hypertensive heart failure. All had normal LV ejection fraction. Cardiac amyloidosis had been ruled out in the hypertensive heart failure cohort by using DPD scintigraphy and hematological blood workup. Aortic stenosis diagnosis was verified with echocardiographic measurement of the aortic valve area (AVA). Preoperative data was collected exclusively from patients with severe AS (AVA < 1 cm^2^). HCM diagnosis was based on exclusion criteria and an interventricular septum hypertrophy (IVST > 14 mm) not explained by increased afterload or infiltrative cardiac disease.

### 2.3. Validation Cohort Population

In total, 20 patients taken from a local data base at Karolinska University hospital with verified ATTR-CA (ATTRwt = 17 and ATTRv = 3) validated the cut off values from deviation study testing RWT and RWT/SaVR.

### 2.4. Data Collection

All clinical records were collected from a database at Umea University, within a catchment area of the northern hospital region in Sweden. The following information was collected from all patients included in this study: mortality, height, weight, NT-pro BNP, Troponin T, ECG, and echocardiography. TTR-gene sequencing results were collected from the ATTRv group. DPD-scintigraphy Perugini gradings were collected from the ATTRwt group.

### 2.5. Electrocardiography

A standard 12-lead ECG examination (50 mm/s, 0.1 mV/mm) was recorded for all patients at the time, for diagnosis. In this study we used the S-wave amplitude in aVR (SaVR), more suited for international standards, in contrast to our previous study utilizing the inverted equivalent, R-wave amplitude in -aVR [14]. ECG amplitudes were excluded from analysis if a deviation from the normal electrical conduction occurred due to ventricular pacing or left bundle branch block. All ECG amplitudes were measured manually and presented in mV.

### 2.6. Echocardiography

Echocardiographic examination was performed using a Vivid E9 system (GE Medical Systems) equipped with an adult 1.5–4.3 MHz phased array transducer. All echoes were performed and analyzed by one investigator (co-author PL). Standard views were used from the parasternal long axis, short axis, and the apical four-chamber views. Chamber quantification and flow velocities were obtained using pulsed and continuous-wave Doppler techniques, as proposed by recent guidelines [18,19].

Relative wall thickness (RWT) was calculated according to the American society of echocardiography (ASE) recommendations (2 × posterior wall thickness (PWT)/Left ventricular diastolic diameter (LVDd)). LV mass was calculated through the Devereux formula (0.8(1.04([LVDd + PWT + IVSd]^3^ [LVDd]^3^)) + 0.6). LV wall thickness was measured as septal thickness + posterior wall thickness. Pulsed wave Doppler analysis was also undertaken, to assess the transmitral early diastolic E velocity and E deceleration time [20].

All Doppler recordings were obtained at a sweep speed of 50–100 mm/s with a superimposed ECG (lead II). Off-line analyses were completed using commercially available software (General Electric, EchoPac version BT 13, 113.0 Waukesha, WI, USA), and the means of three consecutive cardiac cycles were calculated.

### 2.7. Assessment of LV Deformation Function, LV GLS

Anatomical landmarks were used, and care was taken with echocardiographic image acquisition to ensure adequate LV tracking, and to avoid foreshortening of the LV cavity when measuring the global strain of the LV. Longitudinal myocardial deformation was assessed by two-dimensional echocardiography using speckle tracking and was analyzed offline. From the apical four-chamber, two-chamber, and apical parasternal long-axis views, the endocardial border of the septal, apical, and lateral wall of the LV were undertaken manually in order to analyze global LV strain measurements. Strain recordings from three cardiac cycles were averaged to assess the global longitudinal strain (GLS). GLS was measured at end systole with the reference point set at the onset of two consecutive QRS-complexes of the superimposed ECG. We also calculated relative apical sparing (RELAPS) as the average apical strain/(average basal strain + average mid strain). Strain analyses were measured using a dedicated workstation (General Electric, EchoPac version BT 13, 113.0, Waukesha, WI, USA).

### 2.8. DPD-Scintigraphy

All patients were investigated with an Infinia Hawkeye hybrid single-photon-emission computed-tomography gamma camera (SPECT-CT; General Electric Medical Systems) with a low-energy, high-resolution collimator. An intravenous injection of ~750 MBq DPD was performed 3 h prior to the acquisition of a whole-body planar image, followed by a non-contrast, low-dose CT scan and a SPECT acquisition, which provided 60 projections, iteratively reconstructed into a 128 × 128 matrix (OSEM, 3 iterations, 10 subsets) with scatter and CT-based attenuation correction. Reconstruction of SPECT images was performed on the Xeleris workstation (GE Healthcare, Chicago, IL, USA). DPD scores were reported by two experienced clinicians using the Perugini grading system [21], with grade 0 being negative and grades 1–3 increasingly positive.

### 2.9. Statistics

Statistical analyses were performed using SPSS^®^, version 26 (IBM). Data were presented as either mean and standard deviation or median and interquartile range for continuous variables. Normal distribution was tested using Shapiro–Wilk analysis. Percentages were used to describe categorical variables. Categorical variables were compared using Chi-square tests, and continuous variables were compared using Student’s *T*-test or Mann–Whitney non-parametric test. ROC analysis was performed to determine the area under the curve, as well as to find optimal sensitivity, specificity, negative and positive predictive values, and accuracy. A *p*-value of less than 0.05 was considered as of statistical significance.

### 2.10. Ethics

All subjects gave their informed consent for inclusion before they participated in the study. The study was conducted in accordance with the Declaration of Helsinki, and the protocol was approved by the Ethics Committee of Umea (DNR: 2016-435-31M, 2018-137-32M, 2018-418-32M).

## 3. Results

Demographic and echocardiographic data from both ATTR-CA and LVH is presented in Table 1.

### 3.1. Patient Characteristics

ATTR-CA patients were older than LVH patients (mean = 76 years for ATTR-CA and mean = 68 years for LVH. (*p* = 0.001)). HR and height were higher in ATTR, whereas weight was higher in LVH. Troponin-T (*p* = 0.02) differed between groups, whereas NT-proBNP did not. Systolic and diastolic blood pressure were higher in LVH patients (*p* < 0.001) (Table 1).

Significantly increased posterior wall thickness (PWT) (*p* = 0.003), and relative wall thickness (RWT) (*p* < 0.001) were observed in the ATTR-CA group. LVEF/LV mass was significantly lower in ATTR-CA (Table 1).

Furthermore, RWT/SaVR and PWT/SaVR were both significantly increased in the ATTR-CA group compared to the control group (*p* < 0.001).

### 3.2. ROC-Analysis

Four echocardiographic and ECG-derived parameters with a statistically significant difference between groups were further investigated with ROC (receiver operating characteristics) curves. RWT/SaVR generated the highest AUC among the studied selection (AUC 0.95, *p* < 0.001 followed by RWT (AUC 0.849, *p* < 0.001). In comparison, PWT/SaVR had AUC of 0.95, (*p* < 0.001). Remaining parameters produced the following AUC: RELAPS (AUC 0.79, *p* < 0.001) and PWT (AUC 0.84, *p* < 0.001).

RWT/SaVR > 0.7 presented the highest combined sensitivity (97%) and specificity (90%) to identify ATTR-CA (*p* < 0.001), and PWT/SavR (*p* < 0.001) had a sensitivity of 94% and specificity of 82%. RWT > 0.5 had the highest combined sensitivity and specificity, which was 84% and 82% (*p* < 0.001). RELAPS > 1.2 showed the highest combined sensitivity and specificity of 74% and 76% (*p* < 0.001). PWT > 11.5 had sensitivity and specificity of 82% and 78%, respectively (Table 2 and Figure 1).

### 3.3. External Validation of RWT/SaVR

For external validation of the use of RWT/SaVR, the accuracy was investigated randomly in 20 patients diagnosed with ATTR-CA from Karolinska University Hospital with septum > 14 mm. RWT measurement from echocardiography was collected retrospectively from the clinical records at the time of diagnosis. SaVR was measured from the ECG taken at the time of diagnosis. Pacemaker rhythm and LBBB were excluded. All patients had a positive DPD-scintigraphy and AL amyloidosis had been excluded using biomarkers and/or biopsy. Overall, 85% of the patients had RWT > 0.5 and 100% had RWT/SaVR > 0.7.

## 4. Discussion

The results from this validation study highlight the strong diagnostic accuracy of simple, readily and widely available ECG- and echocardiography-based variables in the work up of suspected ATTR-CA. The main finding of this study is the very strong diagnostic accuracy of using RWT as a marker for concentric increased wall thickness and even stronger accuracy using the combination of ratio RWT/SaVR. This confirms results using RWT from earlier studies of patients and controls [14,22].

Relative wall thickness calculated from the equation PWTx2/LVDd, is in itself a useful measurement in ATTR-CA work up as it combines two characteristics of the disease: concentric hypertrophy and reduced LV cavity. LVH-related heart enlargement is mainly centered in the interventricular septum with less involvement of the posterior wall. In comparison, ATTR-CA causes a general and concentric thickening of the heart, including increased RWT. This general build-up of cardiac transthyretin in ATTR-CA also affects the LVEDD in a negative manner, generating smaller diameters, compared to LVH. As part of the concentric increase in wall thickness, amyloid infiltration also involves the atriums [23] and right ventricle [10], as well as the valves [24].

We found a cut-off value using RWT > 0.5 accurately predicting ATTR-CA. This corresponds well to the report from Boldrini, et al., suggesting > 0.6 as a strong predictor for ATTR-CA [22]. It is unclear how RWT was calculated in the study from Boldrini et al., but Another definition of RWT is IVS + PWT/LVDd, with cut-off > 0.6 suggested to be a red flag for ATTR-CA [4].

In addition, we have, in a recent study, also found RWT being predictive for ATTR-CA years before final diagnosis [9].

SaVR represents the ventricular excitation registered from a −150° angle of the heart (lead aVR). Most conditions causing increased cardiac wall thickness are characterized by myocardial hypertrophy and thus show electronically active myocardium causing higher QRS-amplitudes. Myocardial amyloid buildup in the heart, however, has the opposite effect, lowering QRS-amplitudes. The mechanism behind this is most probably amyloid infiltration, which suppresses the electrical signal.

The sensitivity and specificity of RWT/SaVR in our study (97% and 90%) are surprisingly high. They are actually comparable to DPD-scintigraphy (98% and 92%) and higher when compared to both MRI (84% and 87%) and PET (78% and 95%) presented in a recent meta-analysis study [25]. However, the meta-analysis compared the modalities to the reference method, endomyocardial biopsy, and included mixes of both ATTR and AL amyloidosis and, furthermore, did not (as we did) exclude patients with a less pronounced increase in LV wall thickness. Nevertheless, the high accuracy of the RWT/SaVR ratio borders on diagnostic, and confirms that it is a very powerful screening tool for ATTR-CA.

Despite the increasing availability of cardiac MRI, PET, and technetium bone tracer scintigraphy in some parts of the world, these investigations remain costly and not readily accessible for a vast number of patients. The availability of ECG and echocardiography makes the RWT/SaVR ratio a highly useful tool for selecting patients for further work up, and possibly for epidemiological studies where advanced imaging is not possible.

Limitations: There are limitations in this study. We used a cut-off value of >14 mm investigating ATTR-CA whereas others recommended a cut of >12 mm. However, we consider >14 mm to be a more appropriate cut-off due to the numbers to investigate with DPD scintigraphy for final diagnosis. Patients with aortic stenosis and patients with HCM had not undergone work up to rule out CA (DPD scintigraphy). However, mean age in each group (62 and 65 years) reduces the suspicion of ATTRwt.

Being a retrospective study, the analyses were not blinded and therefore at risk of bias. Furthermore, the study does not include patients with very mild disease nor patients with AL amyloidosis, so further evaluations including a more heterogenous CA population, and preferably more centers, are warranted.

## 5. Conclusions

We confirm the very strong diagnostic accuracy of using RWT/SaVR to identify ATTR-CA in patients with moderate or advanced disease. RWT/SaVR > 0.7 demonstrated the highest combined sensitivity and specificity, with which to identify ATTR-CA, when compared with conventional echocardiographic variables. The proposed method is simple to measure and widely used in cardiology centers.

## Figures and Tables

**Figure 1 jcm-11-04120-f001:**
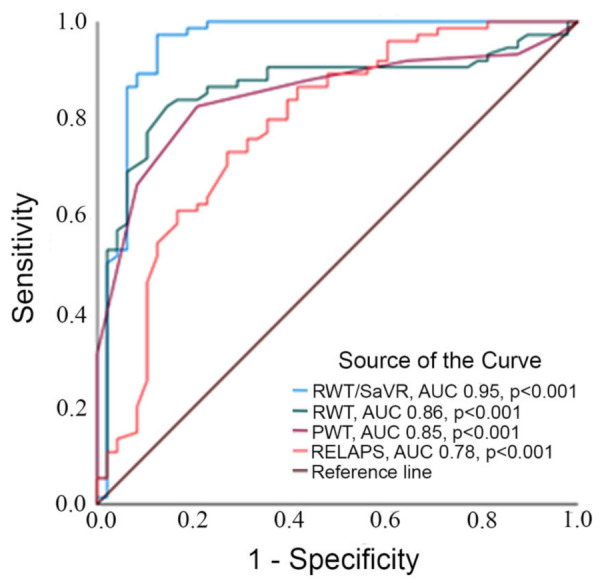
Legend to figure: ROC curve analyzing area under the curve testing RWT/SaVR, RWT, PWT and RELAPS.

**Table 1 jcm-11-04120-t001:** Data shown presented as mean ± SD or median (IQR) (Median(IQR) in Italic style). HR = heart rate, IVSDD = interventricular septal diameter diastole, LVDD = left ventricular diastolic diameter, LVEF = left ventricular ejection fraction, PWT = posterior wall thickness, RWT = relative wall thickness, RELAPS, relative apical sparing, GLS = global longitudinal strain, LAVI = left atrial volume index, LVMI = left ventricular mass index, E = early diastole, and DT = deceleration time.

		ATTR-CA		LVH	
	N	Mean/*Median* SD/*IQR*	N	Mean/*Median* SD/*IQR*	*p*-Value
Age (years)	102	76 ± 8.3	65	68 ± 12.7	0.001
HR (bpm)	101	71 ± 12.2	64	66 ± 13.1	0.039
Height (cm)	99	175 ± 7.7	63	174 ± 10.9	0.005
Weight (kg)	101	76 ± 14.6	63	84 ± 17.9	0.049
Systolic blood pressure, mmHg	98	130 ± 18	64	142 ± 20	<0.001
Diastolic blood pressure, mmHg	96	77 ± 10	64	83 ± 11	<0.001
Log NT-proBNP, ng/L	*93*	*3.1 (1.0)*	*48*	*3.0 (2.5)*	*0.18*
Troponin-T, ng/L	*77*	*30 (33)*	*36*	*21 (26)*	*0.02*
IVSDD (mm)	102	18.7 ± 3.3	65	17.6 ± 3.1	0.294
LVDD (mm)	102	43.9 ± 5.5	65	48.3 ± 6.5	0.121
LVEF	102	54 ± 11	65	57 ± 12	0.436
PWT (mm)	*102*	*13.5 (2.7)*	*65*	*10.2 (1.6)*	*0.003*
PWT/SaVR	81	47 ± 39	55	12 ± 6	<0.001
RWT (mm)	*102*	*0.61 (0.22)*	*65*	*0.43 (0.11)*	*0.000*
RWT/SaVR	81	2.29 ± 1.87	54	0.62 ± 1.23	<0.001
RELAPS	*92*	*2.0 (1.2)*	*57*	*0.8 (0.5)*	*<0.001*
GLS,%	91	−14.1 ± 5.0	58	−13.5 ± 4.0	0.388
LAVI, ml/m^2^	*99*	*39 (17)*	*51*	*39 (20)*	*0.225*
LVMI, g/m^2^	99	189 ± 50	63	170 ± 50	0.187
LVEF/LV mass	99	0.30 ± 0.10	63	0.35 ± 0.12	0.004
E velocity, cm/s	97	62 ± 36	63	63 ± 40	0.899
E DT, ms	90	184 ± 77	61	200 ± 85	0.241

**Table 2 jcm-11-04120-t002:** RWT = relative wall thickness, RELAPS = relative apical sparing, and PWT = posterior wall thickness.

	AUC	Cut-Off Value	Sensitivity [%]	Specificity [%]	PPV	NPV	Accuracy	*p*-Value
RWT/SaVR	0.95	0.7	97	90	90	92	91	0.000
RWT	0.85	0.5	84	82	94	72	83	0.000
RELAPS	0.79	1.2	74	76	82	63	73	0.000
PWT, mm	0.84	11.5	82	78	88	75	82	0.000

## Data Availability

The data presented in this study are available on request from the corresponding author. The data are not publicly available due to ongoing research analysis.

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
