# Peer review of "RWT/SaVR—A Simple and Highly Accurate Measure Screening for Transthyretin Cardiac Amyloidosis"

_jcm, 2022, doi:10.3390/jcm11144120_

Round 1

Reviewer 2 Report

The work is well written. The authors identify a new parameter capable of differentiating  cardiac amyloidosis from other forms of left ventricular hypertrophy. Unfortunately I think that this parameter (RWT/SaVR) is surpassed by other parameters  assessed  by cardiac magnetic resonance in other recently published studies. Parameters such as LVEF / LV mass ratio, LVEF / GLS ratio have recently been evaluated in identifying patients with cardiac amyloidosis. The authors could cite these articles. Authors  could also mention the recently proposed multiparametric echocardiographic score (IWT score) in the diagnosis of cardiac amyloidosis. Furthermore, it would be advisable to increase the limits section of the study specifying why other echocardiographic parameters already validated to differentiate cardiac amyloidosis from other types of hypertrophy were not evaluated.

Round 2

Reviewer 1 Report

Thank you for your prompt revision according to the comments. Although some limitations such as either RWT/SaVR has the same discriminating power in the patients with lesser LV wall thickness (<14 mm) are still remained but your work could give a clinical message by suggesting a simplified tool for screening the patients with thickened LV wall. For some of the limitations mentioned in the first review, please let us know the results in a good paper through additional research currently in progress.

Reviewer 2 Report

no further comments